# Adaptive Responses of the Sea Anemone *Heteractis crispa* to the Interaction of Acidification and Global Warming

**DOI:** 10.3390/ani12172259

**Published:** 2022-08-31

**Authors:** Yangyang Wu, Wenfei Tian, Chunxing Chen, Quanqing Ye, Liu Yang, Jiaoyun Jiang

**Affiliations:** 1Key Laboratory of Ecology of Rare and Endangered Species and Environmental Protection, Guangxi Normal University, Ministry of Education, Guilin 541004, China; 2College of Life Sciences, Guangxi Normal University, Guilin 541004, China; 3College of Biotechnology, Guilin Medical University, Guilin 541004, China

**Keywords:** *Heteractis crispa*, Symbiodiniaceae, *p*CO_2_, heat stress, cholesterol

## Abstract

**Simple Summary:**

This study investigated the effects of the interaction of acidification and warming on the photosynthetic apparatus and sterol metabolism of sea anemone *Heteractis crispa*. Thermal stress is the dominant driver of the deteriorating health of *H. crispa*, which might be relatively insensitive to the impact of ocean acidification; upregulation of chlorophyll content is suggested as an important strategy for symbionts to adapt to high *p*CO_2_. However, warming and acidification (alone or combined) significantly affected the cholesterol or sterol levels. Indeed, environmental changes like warming and acidification will affect the sterol metabolism and health of *H. crispa* in the coming decades.

**Abstract:**

Ocean acidification and warming are two of the most important threats to the existence of marine organisms and are predicted to co-occur in oceans. The present work evaluated the effects of acidification (AC: 24 ± 0.1 °C and 900 μatm CO_2_), warming (WC: 30 ± 0.1 °C and 450 μatm CO_2_), and their combination (CC: 30 ± 0.1 °C and 900 μatm CO_2_) on the sea anemone, *Heteractis crispa*, from the aspects of photosynthetic apparatus (maximum quantum yield of photosystem II (PS II), chlorophyll level, and Symbiodiniaceae density) and sterol metabolism (cholesterol content and total sterol content). In a 15-day experiment, acidification alone had no apparent effect on the photosynthetic apparatus, but did affect sterol levels. Upregulation of their chlorophyll level is an important strategy for symbionts to adapt to high partial pressure of CO_2_ (*p*CO_2_)_._ However, after warming stress, the benefits of high *p*CO_2_ had little effect on stress tolerance in *H. crispa*. Indeed, thermal stress was the dominant driver of the deteriorating health of *H. crispa*. Cholesterol and total sterol contents were significantly affected by all three stress conditions, although there was no significant change in the AC group on day 3. Thus, cholesterol or sterol levels could be used as important indicators to evaluate the impact of climate change on cnidarians. Our findings suggest that *H. crispa* might be relatively insensitive to the impact of ocean acidification, whereas increased temperature in the future ocean might impair viability of *H. crispa*.

## 1. Introduction

As a consequence of human activities and the burning of fossil fuels, atmospheric carbon dioxide (CO_2_) has increased to its highest level in past decades [1], with 30% of the CO_2_ released into the atmosphere being absorbed by the oceans [2]. This has disturbed seawater carbonate chemistry and will increase acidity by 0.3–0.4 pH units by the end of the 21st century [3]. Meanwhile, increased anthropogenic activities have promoted emissions of greenhouse gases (especially CO_2_) [4]. Simulations by the Intergovernmental Panel on Climate Change (IPCC) have indicated a 0.3–4.8 °C increase in the average global surface temperature of the ocean by the end of the 21st century [5]. Marine ecosystems are very sensitive to climate change [6], which is happening at an unprecedented pace that may induce adverse effects on marine organisms [7]. In addition to risks posed by climate change, marine ecosystems are exposed to a suite of local pollutants, including chemical pollutants, petroleum hydrocarbons, and persistent organic pollutants (POPs) [8,9,10,11]. Indeed, the combination of climate change stressors (warming, acidification, high ultraviolet radiation, etc.) and local pollutants can increase threats and risks to marine ecosystems [11].

Coral reefs are among the world’s most biologically complex ecosystems, with high societal and economic values; however, they are in significant decline globally due, in part, to rapid climatic changes [12]. Coral cover has declined by approximately 1–5% per year [13], and 70% of all tropical reefs could disappear by 2050 [14]. Ocean acidification and global warming are two of the most important threats to the existence of coral reefs. Studies of the effects of acidification on corals have commonly focused on calcification [15,16,17,18,19], growth [20,21,22], physiology [23,24,25], and phenotype [26,27], with only a few studies reporting that acidification is associated with coral bleaching [28,29]. Indeed, a large body of literature has demonstrated that global warming affects the coral community [30,31], and its health [32,33,34], metabolism [33,35,36,37], and calcification [38,39].

However, acidification and warming are predicted to co-occur in oceans [3], and the combined effects of acidification and global warming might lead to a significant decrease in coral diversity [25]. Understanding the combined ecological risks of acidification and warming on coral populations is urgently required to predict potential shifts in coral community structure under future global climate change. Accumulating evidence has shown that complicated interactions between these two stressors would have deleterious effects on coral ecosystems [18,25,40,41]. Indeed, the combination of warming and acidification has a negative effect on coral calcification [41], which likely leads to severely reduced coral diversity [18,25,40]. In addition, the synergistic effects of global warming and acidification are highly variable among and within taxa [18,28,34,41,42].

Corals and sea anemones belong to the class of anthozoans [43]. Reef-building corals are less-studied laboratory subjects because they grow slowly and are difficult and costly to maintain. Compared to most corals, sea anemones thrive under a wider array of conditions in the laboratory. Most importantly, their process of endosymbiosis establishment is similar to that of many reef-building corals, highlighting the relevance of the sea anemone as a model organism. The sea anemone, *Heteractis crispa*, exists widely throughout the tropical and subtropical waters of the Indo-Pacific region, from the eastern coasts of Africa to Polynesia and from southern Japan to Australia and New Caledonia [44], providing a habitat for 14 of the 30 species of anemonefishes (also known as clownfish) [45]. Moreover, *H. crispa* requires low feeding levels for maximal growth and is highly dependent on nutrients provided by dinoflagellate endosymbionts (Symbiodiniaceae) [46,47,48]. This characteristic is advantageous to studying the interaction between cnidarian hosts and their symbionts in this system. However, there are no data available regarding the responses of *H. crispa* to the interaction of acidification and global warming.

In this study, we investigated how the individual and combined effects of acidification and warming might affect the phenotype (Symbiodiniaceae density, total chlorophyll, and maximum quantum yield of photosystem II (PS II; Fv/Fm)) of *H. crispa*. Sterols are vital structural and regulatory components in eukaryotic cells; our previous study found that sterols, especially cholesterol, play an important role in the bleaching process of *H. crispa* [33]. Therefore, we used cholesterol and total sterol as important biomolecules to evaluate the sterol metabolic response of *H. crispa* to climate change [33,49].

## 2. Materials and Methods

### 2.1. Growth Conditions, Acclimation, and Treatments

Sea anemones (*H. crispa*) were obtained from the Aquarium of Yunlong Town (Haikou, China). The growth conditions and acclimation methods of *H. crispa* were similar to those in our previous study [33]. In brief, sea anemones were maintained in artificial 35‰ seawater in a recirculation aquaculture system (120 × 50 × 60 cm) under controlled conditions (24 ± 0.1 °C, 16-h/8-h light/dark cycles with approximately 100 µmol·photons·m^−2^·s^−1^ light during light phases). Seawater was prepared using sea salt (Reef crystals, Lorraine, France) and filtered through 1 µm pore size filters. Healthy cultures were transferred into fresh media and acclimated for at least half a month before the experiments were performed. Animals were fed with brine shrimp weekly and then starved for approximately 7 days before the experiments to avoid sample contamination by food metabolites. Sea anemones that were not subjected to stresses served as controls.

To investigate the interactive effects of acidification and warming on *H. crispa*, we experimentally manipulated the partial pressure of carbon dioxide (*p*CO_2_) and temperature within indoor mesocosms. Increased CO_2_ and temperature levels were chosen based on the IPCC scenario WG RCP 8.5 [5]. Specifically, conditions used were control conditions (Control: 24 ± 0.1 °C and 450 μatm CO_2_); acidification conditions (AC: 24 ± 0.1 °C and 900 μatm CO_2_); warming conditions (WC: 30 ± 0.1 °C and 450 μatm CO_2_); and combined conditions (CC: 30 ± 0.1 °C and 900 μatm CO_2_). Briefly, after acclimation, three identical aquariums (three individuals were added to each aquarium) were used for each treatment group. Note that because of limited space, the four treatments did not run concurrently, although we carried out a control treatment and a stress treatment simultaneously each time; thus, the entire research comprised three consecutive trials. In total, 54 *H. crispa* individuals were measured in this study. Air was bubbled into the control and WC group, while CO_2_ mixed with air was bubbled into the AC and CC groups, which was controlled using an air and CO_2_ gas flow adjustment system. The temperature of the seawater was controlled using 300 W submerged heaters (E300, EHEIM, Deizisau, Germany). During the experimental period, the seawater temperature and pH for each treatment group were measured twice a day (Appendix A). Salinity was measured every day using a multiparameter sensor YSI (YSI, Yellow Springs, OH, USA). Total alkalinity was determined using potentiometric titration (888 Titrando, Metrohm, Riverview, FL, USA) [50] and the other carbonate parameters were calculated using CO2SYS software [51] (Appendix A).

### 2.2. Tissue Collection and Physiological Measurements

Sea anemone tentacle samples were collected on days 1, 3, 6, 9, and 15 to determine the maximum quantum yield of PS II (Fv/Fm), Symbiodiniaceae density, and the chlorophyll concentration. The sample collection protocol was conducted as described in previous reports [34,52]. Briefly, 10 tentacles were removed from each animal, and parts of the tentacle (1 cm long) were used for Fv/Fm measurement. The remaining parts were cut lengthwise, scraped to separate the endodermal cell layer, which contains the Symbiodiniaceae, from the epidermal tissue, which were then placed into 5 mL centrifuge tubes containing 3 mL of 0.22 μm-filtered, chilled ddH_2_O at 4 °C [33]. The tissues were homogenized for 1 min using a saw-tooth homogenizer. The homogenate was then centrifuged at 5000× *g* for 15 min to separate the Symbiodiniaceae from the host. The supernatant (host fraction) was used for protein determination according to a previously described method [53]. The pellet was vortexed for 1 min and resuspended in 4 mL of chilled ddH_2_O. Next, 500 µL of the resuspended pellet was used to measure symbiont cell density and another 500 µL was used to measure the chlorophyll concentration. The remaining 3 mL of solution was recentrifuged at 13,000× *g* for 3 min to collect the Symbiodiniaceae cells and stored at −80 °C for subsequent measurement of the sterol content.

Fv/Fm was measured using a chlorophyll fluorimeter (WALZ, Effeltrich, Germany), with a measuring intensity of 4, a saturating intensity of 7, and a gain and damping of 2 [54]. The Symbiodiniaceae density was normalized to the host protein mass [53] and was determined using a 0.1 mm deep Improved Neubauer Hemocytometer (ThermoFisher Scientific, Waltham, MA, USA) by measuring the cell number at defined intervals. Pigment extraction and measurement were performed according to previously published methods [54]. Methanol was used as the extraction solvent (1 mL for each sample), and extraction was carried out at 60 °C in the dark for 30 min. Extracts were then centrifuged at 5000× *g* for 15 min to remove algal debris. Pigment absorbances were determined using a spectrophotometer (GeneQuant 1300, Biochrom, Holliston, MA, USA) using light wavelengths of 664 nm and 630 nm for chlorophyll (chl) a and chl c, corrected by subtracting the absorbance value (Abs) at 750 nm. Equations used to determine the pigment concentrations were as follows:chl a = 11.85 × [(Abs 664 nm − Abs 750 nm) − 0.08 × (Abs 630 nm − Abs 750 nm)]; (1)
chl c = 24.52 × [(Abs 630 nm − Abs 750 nm) − 1.67 × (Abs 664 nm − Abs 750 nm).(2)

### 2.3. Sterol Extraction and Analysis

We used gas chromatography mass spectrometry (GC/MS) analysis to determine the total sterol content and cholesterol content on day 3 and day 15, [33,34]. Before sterol extraction, Symbiodiniaceae samples were lyophilized. Sterols were extracted and analyzed according to our previous study [33]. Briefly, a 1 µL portion of each derivatized sample was injected into a GC/MS single quadrupole (GC/MSD) system. The collected data were analyzed using Agilent GC/MSD Productivity ChemStation and AMDIS (Automated Mass spectral Deconvolution and Identification System) software (Agilent, Santa Clara, CA, USA).

### 2.4. Statistical Analysis

All treatments were replicated at least three times. For all tested parameters, data are represented as the means ± standard deviation (SD). The data’s normality and homogeneity of variance were tested using Shapiro–Wilks and Levene’s tests, respectively. The data showed homogeneity and normal distribution, and were analyzed using analysis of variance (ANOVA). A statistically significant difference was accepted at *p* < 0.05. In the figures, asterisks (*) indicate *p* values of 0.05 or less.

## 3. Results

### 3.1. Maximum Quantum Yield of Photosystem II and Chlorophyll

The maximum quantum yield of photosystem II (Fv/Fm) differed among the different treatments (Figure 1A). Acidification alone had no apparent deleterious effect on the photochemical efficiency of photosystem II (*p* > 0.05); indeed, we observed an upward trend in Fv/Fm values over the first 3 days, although the difference was not significant (*p* > 0.05) (Figure 1A). Similar results were observed in the CC group during the first 3 days. However, after 3 days of exposure in the CC group and 2 days in the WC group, the Fv/Fm values of both treatment groups decreased significantly (*p <* 0.05) (Figure 1A).

Similar to the results observed for Fv/Fm, on day 3 we detected that the level of chlorophyll was significantly higher in the AC group compared with that in the control (*p <* 0.05) (Figure 1B and Appendix A). However, total chlorophyll was negatively influenced by warming, being generally lower in the WC and CC groups, with a decrease starting on day 3 and day 6, respectively (*p <* 0.05) (Figure 1B). No pigment bleaching was observed during the experiment in the AC and CC groups, but was observed in the WC group (Figure 1C).

### 3.2. Symbiodiniaceae Density

Consistent with the results of the chlorophyll content and Fv/Fm value, acidification alone appeared to have little effect on Symbiodiniaceae density (*p* > 0.05) (Figure 2). However, the density of symbionts decreased significantly (*p <* 0.05) after 3 days under both high temperature stress and combined stress (Figure 2). Apparently, warming alone had a greater effect on Symbiodiniaceae density than the combined treatment (*p <* 0.01).

### 3.3. Cholesterol Contents and the Phenotype

Results showed that sterol and cholesterol levels decreased from day 3 and were even lower on day 15 (*p <* 0.05) under CC and WC treatments (Figure 3A,B; Appendix A). Although there was no significant change in sterol metabolism in the AC group on day 3 (*p* > 0.05), it was significantly reduced at the end of the experiment (*p* < 0.05) (Figure 3A,B; Appendix A). In agreement with these findings, we found that the health of *H. crispa* was affected (Figure 3C) by acidification stress, although the photosynthetic apparatus and density of symbionts did not change significantly (Figure 1 and Figure 2). In addition, severe bleaching occurred under WC and CC treatments; the former was more severe, even though we did not detect the color score of the tentacles (Figure 3C).

## 4. Discussion

Coral reefs worldwide have experienced an unprecedented decline over recent decades, primarily because of global climate change [14]. The combined effects of acidification and global warming might lead to a marked decrease in coral diversity [25]. To the best of our knowledge, this was the first study to demonstrate how future ocean conditions will impact *H. crispa*.

In the present study, acidification alone had no apparent effect on the photochemical efficiency of photosystem II. However, we found that on the third day of acidification treatment, there was a slight, but not significant, decrease in Symbiodiniaceae density. Interestingly, we observed an upward trend in Fv/Fm values over the first 3 days, although the difference was not significant. It is possible that the significant increase in the chlorophyll level at this time point was a strategy to compensate for the possible decline in symbiont density [25]. It is suggested that *H. crispa* is able to acclimatize to high *p*CO_2_ environments [18,25,34,55]. In contrast, the chlorophyll content, symbiont cell density, and Fv/Fm value were lower than the control under both warming and combined stress conditions. In addition, it was obvious that the deleterious effects of warming alone on *H. crispa* were more serious than those of the combined treatment. In contrast, a recent study showed that combined stress had a more negative effect on coral health [41]. The reason for this difference might be the mentioned focused on calcification of a scleractinian coral. Data have shown that both warming and acidification have adverse effects on coral calcification [18,38,39,41]. However, the sea anemone used in this study does not have a calcium carbonate skeleton. This further illustrates that *H. crispa* is able to acclimatize to high *p*CO_2_ environments, and that increasing chlorophyll levels is an important strategy by which symbionts adapt to high *p*CO_2_. However, upregulation of chlorophyll levels had little effect on the environmental adaption of *H. crispa* after warming stress. Therefore, the combined effects of acidification and global warming might lead to severe destruction of *H. crispa*, despite the stimulating effect of acidification alone on *H. crispa’s* primary productivity [25,40].

After 15 days of exposure, *p*CO_2_ did not cause a significant bleaching response in *H. crispa* (Figure 3C), as found for the sea anemones *Anemonia viridis* [55] and *Entacmaea quadricolor* [34], which suggested that *H. crispa* might be relatively insensitive to the impact of ocean acidification [34,55]. However, warming, alone or in combination with acidification, caused severe bleaching by the end of the experiment. In addition, warming alone was more harmful to sea anemones than the combined treatment, which supports the view that thermal stress is the dominant driver of deteriorating health [56]. This also supported the view that the benefits of acidification had a little effect on coral survival after thermal stress because of bleaching [57]. Generally, warming, alone or in combination with acidification, causes more serious damage to *H. crispa* than acidification alone [18,34,58,59]. 

Sterols are vital membrane components in all eukaryotic cells [60]. In corals, it has been suggested that sterols are originally synthesized by the symbiotic Symbiodiaceae and not by the host corals [33,61]. Indeed, sterols play an important role in the bleaching process of cnidarians [33,49]. In particular, a lack of cholesterol is a significant feature of sea anemone bleaching [33]. In the present study, we used the cholesterol content and the total sterol content as important indicators and explored the effects of different stresses on the health of *H. crispa*. The results showed that acidification alone significantly affected the cholesterol and total sterol levels. As expected, sterol metabolism was also seriously affected by warming and the combination of acidification and warming. These results imply that although no significant changes in the photosynthetic apparatus were observed during acidification alone, the cholesterol and total sterol contents were affected significantly. Indeed, integration of metabolite and physiological data has a high predictive power to define organism performance [54,62]. In general, cholesterol or sterol is essential to the healthy growth of *H. crispa* and could be used as an important indicator in the scientific evaluation of the impact of climate change on cnidarians [33,49].

## 5. Conclusions

Acidification alone had no apparent effect on the maximum quantum yield of PS II and Symbiodiniaceae density, but did affect sterol levels. Upregulation of the chlorophyll content is suggested as an important strategy for symbionts to adapt to high *p*CO_2_. However, after warming stress, the benefits of high *p*CO_2_ had little effect on stress tolerance in *H. crispa*. Indeed, thermal stress is the dominant driver of deteriorating health of *H. crispa*. The cholesterol and total sterol contents could be used as important indicators to evaluate the impact of climate change on cnidarians. These findings expand our understanding of the responses of marine species to climate change. More importantly, they provide a scientific basis to predict the impact of environmental changes on corals.

## Figures and Tables

**Figure 1 animals-12-02259-f001:**
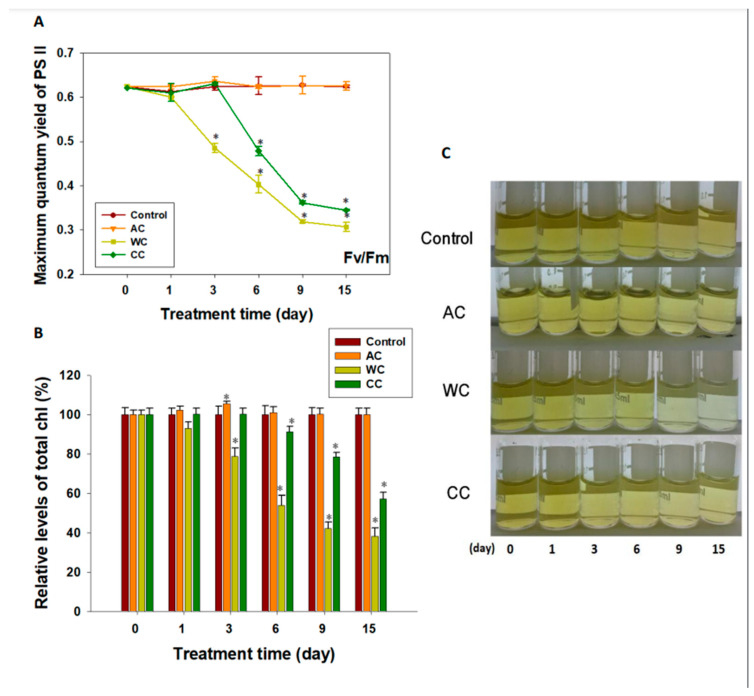
Consequences of different stresses on the photosynthetic activity and apparatus of the symbionts in *H. crispa*. (**A**) Maximum quantum yield of photosystem II (PSII) (*n* = 3); (**B**) Total chlorophyll content (*n* = 3); (**C**) Crude chlorophyll extract. AC, acidification conditions; WC, warming conditions; CC, combined conditions. Asterisks (*) indicate *p*-values < 0.05.

**Figure 2 animals-12-02259-f002:**
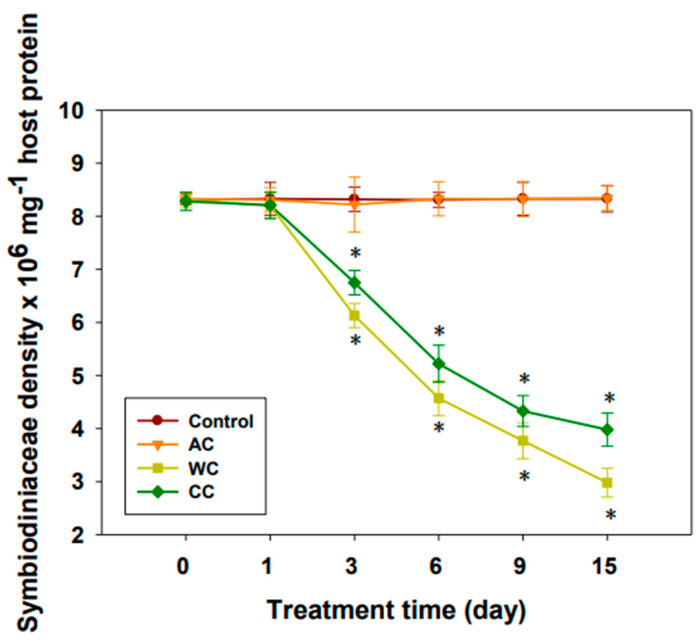
Consequences of different stresses on Symbiodiniaceae density (*n* = 3). AC, acidification conditions; WC, warming conditions; CC, combined conditions. Asterisks (*) indicate *p*-values < 0.05.

**Figure 3 animals-12-02259-f003:**
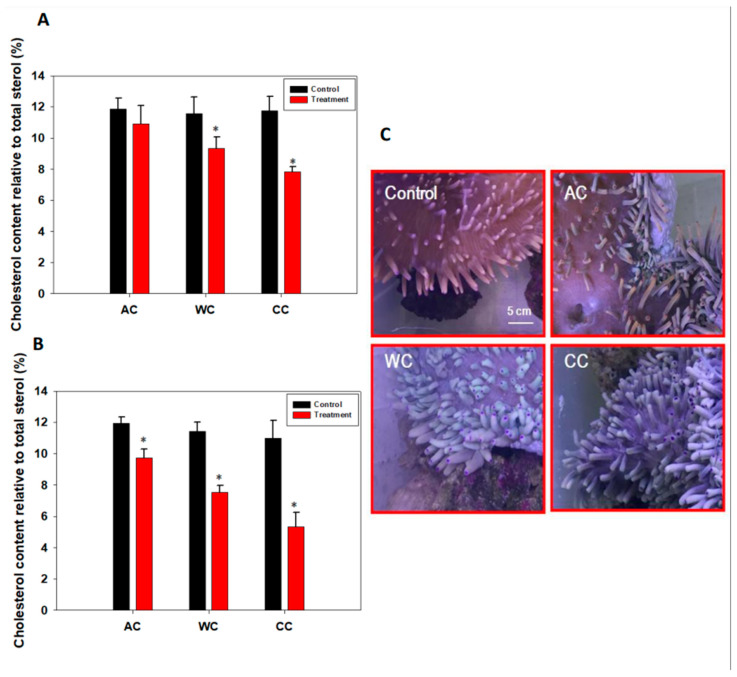
Changes in levels of symbiont cholesterol and the phenotype of *H. crispa*. in response to stress. (**A**) Changes in levels of symbiont cholesterol on day 3 (*n* = 3); (**B**) Changes in levels of symbiont cholesterol on day 15 (*n* = 3); (**C**) Phenotype of *H. crispa* in response to stress. AC, acidification conditions; WC, warming conditions; CC, combined conditions. Asterisks (*) indicate *p*-values < 0.05.

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
