# Peer review of "Adaptive Responses of the Sea Anemone Heteractis crispa to the Interaction of Acidification and Global Warming"

_animals, 2022, doi:10.3390/ani12172259_

Round 1

Reviewer 1 Report

Review

Paper title: Adaptive responses of sea anemone Heteractis crispa to the interaction of acidification and global warming

The authors studied the effects of different rearing conditions on the photosynthetic apparatus and sterol metabolism in the sea anemone Heteractis crispa. They found that acidification conditions (900 μatm CO2) did not affect the photosynthetic apparatus, but demonstrated some effects on the sterol levels. Warming conditions had more expressed effects on the studied parameters. The authors concluded that cholesterol or sterol levels are of great importance in terms of indication of climate change. These findings may have important implications for the management and conservation of benthic communities.

All these reasons explain the relevance of the paper by Yangyang Wu and co-authors submitted to "Animals".

General scores.

The data presented by the authors are original and significant. The study is correctly designed and the authors used appropriate sampling methods. In general, statistical analyses are performed with good technical standards. The authors conducted careful work that may attract the attention of a wide range of specialists focused on anemone ecology.

Suggestions.

The authors should include a "Simple Summary" because this section is mandatory for the journal.

Please, include citations for Fig. S1, S2, and S3.

The authors used a parametric test (ANOVA). This approach requires normal distribution and heterogeneity of the data. The authors should provide the methods used to test these assumptions.

Fig. 1. Please, explain what do mean the vertical bars

Specific remarks.

L 2. Consider replacing “sea anemone” with “the sea anemone”

L 2. "Heteractis crispa" should be italicized.

L 11. Consider replacing “many” with “most important”

L 22. Consider replacing “all  the  three” with “all  three”

L 42. Consider replacing “many” with “most important”

L 55. Consider replacing “to lead” with “leads”

L 59. Consider replacing “poor” with “less-studied”

L 60. Consider replacing “Compared with” with “Compared to”

L 70. Consider replacing “there is no data” with “there are no data”

L 113. Consider replacing “15 day” with “15-day”

L 116. Consider replacing “15 day” with “15-day”

L 151. Consider replacing “were” with “were as follows”

L 173. Consider replacing “in WC group” with “in the WC group”

L 178. Consider replacing “the decrease” with “a decrease”

L 194. Consider replacing “little effect” with “a little effect”

L 205. Consider replacing “at day 15” with “on day 15”

L 220. Consider replacing “compared with” with “compared to”

L 229. Consider replacing “unprecedented decline” with “an unprecedented decline”

L 240. Consider replacing “This” with “It is”

L 246. Consider replacing “study” with “the mentioned”

L 246. Consider replacing “scleractinia” with “scleractinian”

L 257. Consider replacing “anemone” with “anemones”

L 263. Consider replacing “little effect” with “a little effect”

L 266. Consider replacing “is has been” with “it has been”

Author Response

Dear reviewer, 

Sincerely
Jiaoyun Jiang

Reviewer 2 Report

There is information on the impact that coral reefs around the world have experienced in recent decades, this study provides evidence of the effects of global warming and acidification. The results explain the decline in coral diversity.
It is recommended to widely discuss the causes that generate acidification and warming in the marine ecosystem, and conclude if public policies currently support the exposed problem.

Author Response

Dear reviewer, 

Sincerely
Jiaoyun Jiang

Reviewer 3 Report

The Paper “Adaptive responses of sea anemone Heteractis crispa to the interaction of acidification  and  global warming” reports important data about the sensitivity of  this model organism against two of the many important threats for marine organisms.

I suggest minor revision:

Introduction

Authors should provide more information about the complexity of the problem related to the climate change. For example data on the possible deleterious effects, in presence of other chemical-physical threats should be reported. (for example see Martino et al., 2021, Aquatic toxicology).

Material and methods

Author reports: “In brief, sea anemones were maintained in artificial 35‰ seawater in a recirculation aquaculture system.”

-          Why wasn't filtered natural sea water used? Please explain. In general, the use of filtered sea water, from the sites where the organisms under study live, would allow to mimic the environmental conditions favorably. The authors, in my opinion, should explain whether a check with natural seawater was done in order to confirm the data.

-           

Apart from the above suggestions, I do not find any objection to giving my favorable opinion for the publication of this work in Animals journal.

Author Response

Dear reviewer, 

Sincerely
Jiaoyun Jiang
